# The Function of Cumulus Cells in Oocyte Growth and Maturation and in Subsequent Ovulation and Fertilization

**DOI:** 10.3390/cells10092292

**Published:** 2021-09-02

**Authors:** Bongkoch Turathum, Er-Meng Gao, Ri-Cheng Chian

**Affiliations:** 1Centre for Reproductive Medicine, Shanghai 10th People Hospital of Tongji University, Shanghai 200072, China; bongkoch@nmu.ac.th; 2Department of Basic Medical Science, Faculty of Medicine Vajira Hospital, Navamindradhiraj University, Bangkok 10300, Thailand; 3Shanghai Clinical College, Anhui Medical University, Hefei 230032, China; gaoermeng1130@163.com

**Keywords:** cumulus cells, expansion, oocyte, embryo, pregnancy

## Abstract

Cumulus cells (CCs) originating from undifferentiated granulosa cells (GCs) differentiate in mural granulosa cells (MGCs) and CCs during antrum formation in the follicle by the distribution of location. CCs are supporting cells of the oocyte that protect the oocyte from the microenvironment, which helps oocyte growth and maturation in the follicles. Bi-directional communications between an oocyte and CCs are necessary for the oocyte for the acquisition of maturation and early embryonic developmental competence following fertilization. Follicle-stimulation hormone (FSH) and luteinizing hormone (LH) surges lead to the synthesis of an extracellular matrix in CCs, and CCs undergo expansion to assist meiotic resumption of the oocyte. The function of CCs is involved in the completion of oocyte meiotic maturation and ovulation, fertilization, and subsequent early embryo development. Therefore, understanding the function of CCs during follicular development may be helpful for predicting oocyte quality and subsequent embryonic development competence, as well as pregnancy outcomes in the field of reproductive medicine and assisted reproductive technology (ART) for infertility treatment.

## 1. Introduction

In a good physical condition of the intrafollicular environment, cumulus cells (CCs) surrounding the oocyte, help the ability of oocyte maturation [1]. Understanding how CCs impact oocyte development and protect the oocyte from harmful systemic diseases is crucial for the management of infertility [2]. Moreover, damage of the CCs under various pathogenesis has the possibility to reduce the rates of fertilization and the chance of pregnancy [3]. Current research has been focused on invasive techniques and involves the analysis of the somatic cells surrounding the oocyte, CCs, to determine the important factors of CCs that can predict the oocyte quality for high rates of maturation, fertilization, embryo development, and pregnancy [1,2,3]. Thus, this review discusses the function of CCs and its factors that have been involved in the quality of oocytes in the term of developmental of the granulosa cells and oocyte, the contribution of fertilization and early embryogenesis, and prediction of pregnancy in assisted reproduction technologies (ARTs).

## 2. The Definition of Cumulus Cells and the Differentiation of Granulosa Cells

CCs are the somatic cells surrounding the oocyte. They play an important role in the growth of the oocyte, meiotic maturation, ovulation, and fertilization in mammalians [4]. CCs originate from undifferentiated granulosa cells (GCs) from primordial to preantral follicles. In the ovary, GCs are the primary cell type that provide the physical support and microenvironment required for the growing oocyte [5]. At the stage of the pre-antra follicles, GCs can be divided into two populations: Mural granulosa cells (MGCs) and CCs. MGCs line the wall of the follicle, while CCs are associated with the oocyte. The corona radiata is the first layer of CCs around the oocyte, and CCs form a pseudostratified communicating epithelium and connect with the oocyte by membrane extensions around the corona cells. During follicular development, the structure of the cumulus oophorus is formed by undifferentiated GCs in the antral follicle. MGCs are located on the wall of the follicle, and CCs are directly in contact with the oocyte (Figure 1).

CCs that are in contact with the oocyte form the cumulus-oocyte complex (COC). This complex allows communication between the oocyte and CCs by directly affecting gene expression and protein synthesis, leading to differentiation and expansion of CCs as well as oocyte maturation. CCs connect to the cytoplasm of the oocyte and penetrate the zona pellucida (ZP) with gap junctions [6]. Following the development from the antral follicle to the pre-ovulatory follicles, CCs that undergo proliferation and the gap-junctions are gradually released from the ooplasm with meiotic maturation of the oocyte.

Apart from anatomical differences in the location of the follicle, CCs and MGCs are also functionally distinct [7]. Normally, the function of CCs is to support the growth of oocyte, produce hyaluronic acid (HA), and undergo CC expansion in response to FSH, while that of MGCs is to carry out the endocrine function(s) and support the development of the follicle [8]. CCs produce HA and become expansive, while MGCs do not synthesize HA without expansion [9]. Following LH surge, the structure of the cumulus oophorus undergoes expansion by accumulating an extensive HA in the extracellular matrix (ECM) that helps ovulation and the subsequent fertilization in the ampulla of the oviducts [10].

The morphological structure and the function may be different when CCs and MGCs are from different sizes of follicles and different phases of menstrual cycles. COCs from small antral follicles without FSH stimulation and LH surge are most likely shown with an undifferentiated form. However, COCs from preovulatory or dominant follicles, especially after LH surge, typically show a differentiated form. At this stage, MGCs and CCs are distinct cells in the follicles. MGCs are compact, dark cytoplasms and shaped as epithelial-like cells (Figure 2a). In contrast, CCs are loose, clear cytoplasms and have a round shape (Figure 2b) when they are stained with alpha-tubulin for observation (unpublished data).

The development of antral follicles is stimulated by gonadotropins and oocyte secreted factors (OSFs). Pre-antral GCs are the common precursors of both MGCs and CCs, and antra formation separates the granulosa cells into MGCs and CCs [7]. The formation of CCs is regulated by OSFs within the follicle, whereas the formation of MGCs is under the regulation of follicle stimulation hormone (FSH) from the outside follicle, i.e., the anterior pituitary gland [11]. FSH is an essential hormone in the follicle for the proliferation and differentiation of granulosa cells. FSH plays roles in the proliferation, growth, and differentiation of GCs by increased vascularization of the theca interna layer of cells peripheral to the basal lamina, the production of follicular fluid, and the differentiation of MGCs and CCs [12].

The differentiation of undifferentiated GCs into MGCs is stimulated by FSH in close interaction with insulin-like growth factors (IGFs) [13,14]. The FSH-induced differentiation of MGCs plays a crucial role in fertility and prepares the female reproductive organs for sperm transportation, fertilization, and early embryonic development [15]. FSH stimulates the luteinizing hormone receptor (LHR) mRNA expression in MGCs. LHR mRNA is a key marker of MGC differentiation [16]. Levels of the mural marker transcript LHR are highest near the basal lamina and lowest in the CCs [17]. As the MGC phenotype, FSH stimulates MGC transcripts involved in steroidogenesis (Cyp11a1), ovulation (LHR), and immune function (Cd34). In bovine GCs, insulin-like growth factor-I (IGF-I) has a role in follicle regulation that acts with FSH to regulate granulosa cell growth. FSH and OSFs are important to determine MGC and CC lineages and drive differential expression of gene characteristics of either MGC or CC phenotypes [18]. 

In contrast, CCs in the preovulatory follicle are spared from the differentiation effects of FSH and insulin-like growth factor (IGF) by OSFs [11]. FSH suppresses CC marker transcripts such as S1c38a3 and the anti-Mullerian hormone (AMH) mRNA levels in COC and stimulates mural transcripts, while the oocyte-stimulated SMAD2/3 signaling promotes elevated levels of cumulus marker transcripts and suppresses mural transcripts in CCs such as LHR through growth differentiation factor 9 (GDF9) and bone morphogenetic protein 15 (BMP15) secreted by the oocyte (Figure 3) [19]. GDF-9 binds to TGFβ type I receptor kinase (ALK5) [20] and bone morphogenetic protein receptor type II (BMPRII) [21], while BMP-15 binds to the receptors ALK6 and BMPRII. GDF-9 and BMP-15 have a key role in the growth, differentiation, and function of GCs and thecal cells during follicular development playing a vital role in the development of oocyte, ovulation, fertilization, and embryonic competence. [22,23]. In mice, it has been reported that the differentiation of CCs required OSFs through the interaction of GDF9 with its receptor and the activation of extracellular signal-regulated kinase ½ (ERK1/2) through epidermal growth factor receptor (EGFR), which cooperatively promoted SMAD2/3-mediated gene transcription [24]. It seemed that human CCs in response to FSH required the IGF1R activity and subsequent protein kinase B (AKT) activation, and this effect of FSH was abolished when the activity of IGF1R was inhibited [13]. In the absence of FSH, the effect of FSH may terminate further CC proliferation before ovulation.

The transcriptomes of CCs and MGCs in mouse large follicles, CCs were enriched in transcripts associated with the metabolism and proliferation of cells, while MGCs were enriched in transcripts involved in cell signaling and differentiation [25]. In humans, the comparisons between transcriptomes of CCs and MGCs showed significant functional differences in these cells. CCs are involved in steroidogenesis compartments, and MGCs are involved in angiogenesis compartments [26]. In mice CCs, expression of SLC38A3 mediated the transportation of specific amino acids from the CCs to the oocyte in support of oocyte growth [27]. Androgen receptor (AR) expression in the follicle is required for fertilization. In mice, deletion of the AR gene led to infertile and CC defects in morphology and differentiation [28]. Previous studies revealed that inhibited SMAD2/3 activation blocked the ability of the oocyte to stimulate the proliferation of CCs [29], and Smad4 gene deletion in GCs showed defectiveness in the differentiation of CCs [30]. Thus, the differentiation of granulosa cells to CCs needs specific factors and suppresses the activation of MGCs development.

## 3. The Interaction of Oocyte and Cumulus Cells

The role of CCs in oocyte growth is to synchronize nuclear and cytoplasm maturation and to regulate meiosis resumption by providing many factors to oocytes such as cyclic adenosine monophosphate [31], cyclic guanosine monophosphate (cGMP) [32], and regulatory molecules [33,34]. The cAMP and cGMP, secondary messengers, are produced by CCs and involved in communication between CCs and oocytes mediated by gap junction. After birth, GCs keep oocytes to arrest at the prophase stage of meiosis I. After puberty, GCs undergo proliferation and differentiation to CCs for supporting nutrition and initiating meiosis resumption [35]. The first step of meiotic resumption occurs by gonadotropins activating the surrounding GCs. In mammals, the actions of LH and FSH on the meiotic resumption of the oocytes mediate an increase in cAMP production by the activation of mitogen-activated protein kinases (MAPKs) [31,36,37], which act on the GCs and promote an increase in the levels of cAMP within the granulosa cell compartment and a decrease in cAMP and cGMP in the oocyte by reducing the permeability of the gap junction between CCs and oocytes, thus inducing the resumption of meiosis as well as the cumulus expansion in preparation for ovulation and fertilization [32,38].

As oocytes cannot produce glucose as an energy substrate, the CCs metabolize glucose into pyruvate and send energy to the oocyte for maturation [18]. Oocyte-derived BMP-15 and GDF-9 are involved in regulating the glycolysis and synthesis of cholesterol, as GCs serve nutrients to the oocyte with pyruvate, lactate, and products resulting from the biosynthetic pathway of cholesterol via gap junctions [39]. Lipid droplets are the main structure to support energy for mammalian oocyte maturation and subsequent preimplantation of embryonic development. In bovine CC metabolites, lipids were mostly detected in CCs and, in general, their concentrations decreased during maturation. The concentrations of various lipid classes, such as phosphatidylcholine and phosphatidylethanolamine, decreased in CCs during maturation suggesting metabolic cooperation in lipid metabolism between the cumulus cells and the oocyte [40]. Perilipin 3 (PLIN 3) is the factor that has the function of lipid formation and regulation. It has been found that lipid droplets are distributed widely in the oocyte and CCs. PLIN3 was highly expressed in CCs of mature oocytes, and lipid droplet accumulation during oocyte maturation was then affected by PLIN3 cross-communication between CCs and the oocyte [41]. Disruption of communication between CCs and the oocyte led to poor developmental potential or infertility [42].

The interactions between oocyte and CCs involve a gap junction and many signaling factors [35]. Gap junctions between CCs and the oocyte are known as transzonal projections (TZPs), thin cytoplasmic projections, which connect CCs to the oocyte and are crucial for normal oocyte development [43]. The gap-junction channel between oocytes and CCs is GJA4 (gap junction protein α 4, also known as connexin 37, Cx37), which is the major isoform of connexins. Gap-junction-associated protein α 1 (GJA1 or connexin 43, Cx43) is the major isoform of connexins between GCs: MGCs–MGCs and MGCs–CCs. Gap junctions transmit nutrients and small molecules such as ions, metabolites, amino acids, and intracellular signaling molecules from GCs to oocytes via CCs [18]. In the GCs, cGMP maintains TZPs during the preovulatory phase. At ovulation, LH binds to LHR in the MGCs and triggers the release of EGFR ligands, which activates ERK/MAPK and leads to actin cytoskeleton reorganization and TZP retraction into the cell body of CCs [44]. It has been reported in mice that gap junctions are necessary for the in vivo and in vitro maturation of oocytes, the presence of CCs during insemination improved fertilization and the subsequent formation of blastocyst, and this improvement was regulated by gap junctions [45]. During the meiosis resumption of oocytes, the level of cAMP in oocyte was decreased by the LH surge. The LH surge leads to active CD44, which participates in the phosphorylation of gap-junction proteins, and the rupture of gap-junctions between CCs and the oocyte causes a decrease in cAMP transfer to the oocyte, thereby triggering meiotic maturation [46].

Moreover, oocytes help provide OSFs for the GC proliferation, differentiation, apoptosis, metabolism, steroidogenesis, and expansion of CCs by GDF9 and BMP15 [24,47]. GDF9 and BMP15 are two members of the transforming growth factor h (TGFh) superfamily that is expressed by oocytes [48]. Relationships between GDF-9 and BMP-15 as GDF-9/BMP-15 heterodimers, known as cumulin, act as regulators of GC and CC function and improve oocyte quality [23]. During oocyte maturation, GDF9 and BMP15 act upon CCs to induce expansion-enabling factors. OSFs and LH also prevent and reduce apoptosis in CCs. In COCs, cumulin binds to its receptor, BMPR-II, to prevent CC death [49]. Therefore, the COC communication network regulation will have physiological implications for oocyte growth, oocyte maturation, ovulation, and subsequent fertilization and embryonic development competencies.

## 4. The Mechanisms of Cumulus Expansion and Ovulation

The cumulus extracellular matrix (ECM) is mainly the formation of HA, pentraxin-3 (PTX3), TNF-stimulated gene-6 (TSG-6, also known as TNFAIP6), and heavy chains (HCs) of serum-derived inter-α-inhibitor proteins [50]. The intercellular matrix between CCs is composed of HA bound to cell surface receptors (CD44 and Rhamm). Cross-linking proteins will stabilize and organize HA into a specific structure. Versican (VCAN), Tsg6, and the inter-alpha-inhibitor HC each bind HA, whereas PTX3 interacts with multiple Tsg6 molecules. VCAN, an extracellular matrix proteoglycan, interacts with integrins and cell surface proteins through its C-terminal region and anchors the matrix of HA to CCs. The protease ADAMTS-1 (a disintegrin and metalloproteinase with thrombospondin motifs) cleaves VCAN in the βGAG, a chondroitin sulfate substituted midsection, a domain generating HA-binding fragments ending with the neoepitope DPEAAE at the C-terminus in this matrix. The versican G1-DPEAAE, known as versikine, functions in the organization of the HA matrix [51]. It possibly modulates the COC matrix structure and function and its role during ovulation [52]. The cumulus expansion depends on the glycosaminoglycan synthesis in HA in the ECM, where it plays a role as a structural component of cumulus expansion and a signal molecule regulating the maturation of oocytes. CCs produce ECM molecules resulting in cumulus expansion, which is essential for ovulation and fertilization and is predictive of oocyte quality [53]. CC layers are sufficient, and the adequate production of HA followed by cumulus expansion is necessary for oocyte maturation [53].

During the transition from pre-antral to antral follicles, oocyte-associated granulosa cells become competent to undergo cumulus expansion. OSFs promote the differentiation of pre-antral GCs to CCs such as BMP15 [47] and GDF9 [54]. In bovine samples, GDF9 and BMP15 act on undifferentiated GCs to promote the formation of antrum-like structures. A lack of GDF9 and BMP15 lead to GC undifferentiation and antral malformation [55]. In CCs, GDF9 has been reported to induce the expression of several genes, including HA synthase 2 (HAS2), cyclooxygenase 2 (COX2; PTGS2), GREM1, and steroidogenic acute regulator protein (STAR) and to repress LHR, which is important for the development of the follicle and cumulus expansion [30,54]. Thus, the regulation of GDF9 and its downstream factors in CCs could predict the quality and health of oocytes. Interestingly, the innate immune response and cytokine production CD34 antigen and the pathogen recognition receptors are members of the Toll-like receptor (TLR) family that induce cumulus expansion. PGE2 and AREG induce the expression of interleukin (I1)-6 in mouse CCs, and I1-6 acts as a regulator of cumulus expansion. In mice and porcine cumulus expansion, SMAD2/3 signaling is activated by OSFs and EGFR [56]. The activation of SMAD2/3 stimulates HA synthesis and proteins involved in the matrix expansion. An FSH-EGFR activates the SMAD2/3 signaling pathway and is involved in the regulation of cumulus expansion and steroidogenesis. FSH enhances EGF-induced tyrosine phosphorylation of EGFR and stimulates specific EGFR-regulating proteins. Moreover, the syntheses of both HA and progesterone are induced by FSH. Following that, SMAD2/3 activation by GDF9/TGFβ affects gonadotropin-induced HA and progesterone synthesis by porcine CCs [56].

OSFs are also required to enable increases in HAS2, prostaglandin-endoperoxide synthase 2 (PTGS2), PTX3, and Tnfaip6 transcripts during cumulus expansion [57]. These transcripts are required for cumulus expansion, as the phenotype of null mutations in PTGS2, PTX3, or Tnfaip6 genes or the inhibition of HAS2 activity severely compromises cumulus expansion [11,42,58]. There are several studies reporting that HAS2 is necessary for the differentiation and expansion of CCs and correlates with early embryogenesis. The production of the PTX3 gene is an ECM protein that interacts with HA in the expanded cumulus matrix [59]. CCs and GCs synthesize TNFAIP6 in the preovulatory follicle [60]. Inter-α-trypsin inhibitor (IαI) and TNFAIP6 are responsible for ECM formation of CCs during expansion by the stabilization of HA chains. Cleavage of IαI to heavy chains HC1 and HC2 is essential for the binding of HCs and HA, and cumulus expansion stabilization [55]. The TNFAIP6 interaction with HA chains is essential for the further stabilization of cumulus expansion. This binding depends on the interaction of TNFAIP6 with PTX3, upregulated by GDF9 and produced during cumulus expansion into ECM [61].

At the ovulatory phase, cumulus expansion and oocyte maturation occur in the pre-ovulatory follicle. The cumulus expansion process requires the presence of cumulus expansion enabling factors secreted by the oocyte. Cumulus expansion is the formation of the HA-rich extracellular matrix by HAS2, Tnfaip6, PTX3, and VCAN [60]. Cyclic nucleotides (cAMP and cGMP) maintain oocyte arrest. High levels of intracellular cAMP are maintained via both oocyte-mediated syntheses and the cAMP influx produced by the surrounding GCs through the connecting gap junctions [62]. During the LH surge, EGFs including amphiregulin (AREG), epiregulin (EREG), BTC, and PTGE2, are produced by GCs and inhibit the production of cGMP and cAMP. This is followed by a drop of cAMP in the oocyte and the resumption of meiotic progression by protein kinase A (PKA) dephosphorylation and the mitosis promoting factor (MPF) activation as well as meiotic maturation and nuclear envelope breakdown (NEBD), and oocyte maturation [63,64]. LH binds to the LHR in MGCs to induce expression of EGF and activates the EGFR, KRAS, and ERK1/2 in MGCs and CCs. Activated ERK1/2 induces the expression of PTGS2, steroidogenic acute regulatory protein (STAR), HAS2, and Tnfaip6 [65]. The LH surge leads to cumulus expansion, which is mediated by EGF-like peptides (EGFLPs) produced by MGCs [8]. LH binds to its receptors on MGCs and stimulates the expression of EGFLPs [66]. AREG, EREG, and BTC act directly on MGCs and CCs that stimulate meiosis resumption, cumulus expansion, and then ovulation [67]. During oocyte maturation, metabolomics changes in bovine CCs have been reported, in that amino acids are the most increased in concentration in CCs, especially serine. It is possible that serine is essential for oxidative stress reduction by channeling depleted serine stores to glutathione synthesis in the oocyte. Serine may be used to generate one-carbon units for nucleotide synthesis, e.g., glycine or α-ketoglutarate [40]. In human CCs from mature oocyte, long non-coding (lncRNAc) RNAs including NEAT1, MALAT1, ANXA2P2, MEG3, IL6STP1, and VIM-AS1 are involved in apoptosis and extracellular matrix-related functions, and are essential for oocyte growth. Thus, lncRNAs expressed in CCs could regulate essential pathways that contribute to human oocyte maturation, fertilization, and embryo development and provide biomarkers of oocyte quality for the development of non-invasive tests to identify embryos with high developmental potential [67].

In addition, FSH induces AREG and EREG gene expression and stimulates the cumulus expansion in pigs [68], cattle [69], and humans [70]. The mechanism of FSH-induced expansion occurs by an increase in the concentration of cAMP in CCs and increased HAS2 gene expression [71]. FSH stimulates cumulus expansion correlation to accumulate glycosaminoglycans in the ECM. In pigs, the action of insulin-like growth factor 1 (IGF1), a known activator of PI3K/AKT signaling, activates the FSH-stimulated synthesis of HA within the expanded ECM by phosphatidylinositol 3-kinase (PI3K)/v-AKT murine thymoma viral oncogene homolog (AKT)- and mitogen-activated kinase 3 and 1 (MAPK3/1)-dependent mechanisms [56]. The activity of PI3K/AKT is essential for gonadotropin-induced CCs expansion in vitro. The HAS2 and PTGS2 expression in CCs is regulated by the existence of an FSH-activated and PKA-independent pathway [72]. Alterations in cumulus expansion are responsible for the disadvantage of reproduction, either as a direct cause or as a reflection of a decline in the functional and structural qualities of the oocyte (Figure 4), so it may destructively affect the movement of the COC ovulation procedure.

## 5. Cumulus Cells with Fertilization

As CCs are included in the gap junction formation, nutrients, and hormonal transportation to oocytes, CCs also relate to the alteration of sperm physiology by inducing changes in sperm, enabling fertilization and enhancing fertility [73]. In CCs, calcium [Ca^2+^] mobilization is involved in the process of oocyte maturation and acrosome reaction. An LH-induced [Ca^2+^] increase in CCs leads to an increase in the cAMP level and activates the MAPK pathway in CCs, which is involved in the expression of EGF-like factors and influences oocyte maturation. During fertilization, the sperm penetrates the cumulus layer and the ZP. The movement of sperm towards the oocyte is selected by the function of CCs as a selective barrier. CCs secrete progesterone, an acrosome reaction inducer activated by cell Surface Receptor NYD-SP8-induced Ca^2+^ mobilization, and play a role in fertilization by controlling the binding of sperm to the zona pellucida [74]. Progesterone induces hyperactivated flagellar movement and acrosome reactions, as well [75]. Previous studies in mice have shown that CCs provide factors affecting sperm functions. Prostaglandins, PGE1, PGE2, and PFG2a, were detected in the incubation medium of COCs. Blocking the biosynthesis of prostaglandins with indomethacin resulted in a decrease in the rate of fertilization [76].

Metalloproteinase is a protease enzyme with a catalytic mechanism. Disintegrin-metalloproteinase 1 (ADAMTS1), matrix metalloproteinase 2 (MMP2), and TIMP metallopeptidase inhibitor 1 (TIMP1) are metalloproteinases involved in fertilization. In mice, ADAMTS1, an extracellular metalloprotease, induces the development of ovarian follicles by ovulatory hormones and is essential for fertility. In Adamts1^−/−^ mice, the rate of ovulation was reduced by 77%, and the rate of fertilization was reduced by 63%. ADAMTS1 is thus required in both processes of ovulation and fertilization in vivo [77]. In infertile women, MMP2, involved in the breakdown of ECM, is more expressed in CCs, and an increased expression of TIMP1, an inhibitory molecule that regulates matrix metalloproteinases (MMPs), and ADAMTS, in cases of reduced ovarian response, and a decreased fertilization rate may be correlated with reduced fertility [78]. GJA1 and serpin peptidase inhibitor clade E (SERPINE 2) represent gene markers potentially associated with the maturation of oocytes, and PRSS35 may be correlated with the ability of oocyte fertilization [79]. The developing acrosome in spermatids contains pituitary adenylate cyclase-activating polypeptide (PACAP). The PACAP type I receptor present in postovulatory CCs binds with PACAP to induce acrosome reactions in mice [80].

Sperm penetration through the cumulus layer and ZP is enhanced by PACAP. In response to PACAP, CCs released a soluble factor to stimulate sperm motility, acrosome reaction, and fertilization [80]. PTX3 plays a role in innate immunity against selected pathogens and fertilization in females. In mice, PTX3 is produced by CCs during the expansion of CCs and localizes in the matrix. In humans, PTX3 is also expressed in CCs. In Ptx3^−/−^ mice, infertility is associated with severe abnormalities of CCs and a failure of oocyte fertilization in vivo. CCs from Ptx3^−/−^ mice synthesize normal amounts of HA but are unable to stabilize the matrix. Thus, PTX3 is a structural component of CCs, and ECM is essential for female fertility [61]. Inter-α-inhibitor (ITIH), a proteoglycan, is required for reproduction in mammals. It is composed of two homologous “heavy chains” (HC1 and HC2). Prior to the ovulation, HCs are transferred onto HA for the formation of HC·HA complexes and an ECM stabilization around the oocyte that is required for fertilization [81]. In mice, ITIH1 and ITIH3 influence the rates of fertilization. These two proteins are required for ECM formation and stabilization [73].

A previous study has revealed that serine protease 35 (PRSS35) was exclusively expressed in the ovary and that mRNA levels were associated with the potential of oocyte fertilization. The mRNA levels of PRSS35 in CCs of fertilized oocytes were significantly higher than the CCs of unfertilized oocytes [79]. The fertilization of denuded oocytes partially restored the cleavage rate, suggesting that CC-secreted factors are important for fertilization, but attachment between oocytes and CCs is required for optimal fertilization and first cleavage [79]. Human leukocyte antigen-G (HLA-G) plays a role in the maturation of oocytes and embryo implantation. In human CCs, it was found that a low expression of HLA-G was associated with a poor quality of oocytes and poor fertilization and reduced the development of embryos [82]. Before fertilization, the exposure of COCs to hydrogen peroxide reduced the rates of cleavage, but it did not result in the death of CCs or oocytes. In contrast, the exposure of denuded oocytes (DOs) to hydrogen peroxide resulted in oocyte death and a complete block of the first cleavage [83]. Thus, the function of CCs is involved in the protection of oocytes against oxidative stress during fertilization.

## 6. Cumulus Cells with Embryonic Development

Embryo selection with a higher potential of implantation has been one of the major challenges in infertility treatments with ARTs. This selection depends on the morphological criteria of the oocytes and embryos, such as growth rate, early cleavage on day-1, degree of fragmentation, and blastocyst formation [84]. Many factors affect oocyte quality and lead to continued embryo implantation failure implications. These factors control the nuclear and cytoplasmic maturation of the oocytes through complex intrafollicular processes [1]. Due to the crucial role of CCs in the follicle development and growth of oocytes, several studies have shown that oocyte quality was determined by CC viability. Hence, several studies correlating cumulus gene expression with oocyte maturity, fertilization, embryonic development, implantation, and pregnancy have been performed [85].

At the cleavage stage, removal of CCs before IVF in bovine samples significantly reduced the cleavage rate (25% for DOs versus 56% for COCs). Thus, the removal of CCs before IVF affected the cleavage rate through the loss of a factor secreted by these cells. This factor is probably progesterone [1]. Previous studies on transcriptomes in human CCs reported that, in 611 differentially expressed genes in CCs from early and non-early cleavage embryos, 24% were overexpressed in the early cleavage in CCs. These genes were involved in several signaling pathways including cell cycle, survival and death signaling, chemokine and cytokine signaling, angiogenesis, and lipid metabolism [85]. Phosphorylation regulates the activity of 7-dehydrocholesterol reductase (DHCR7) and glutathione peroxidase 3 (GPX3) in CCs and is essential for embryonic development. GPX3 is indicated by a hypoxic environment. Hypoxia and ROS in follicular fluid are negatively associated with embryonic development, pregnancy outcome, and a significantly higher incidence of aneuploidy and spindle defects in oocytes [86]. Cholesterol, which is the major component of progesterone and estrogen synthesis, derives from 7-dehydrocholesterol via DHCR7 induction. In COCs, the inhibition of cholesterol synthesis resulted in the reduction of progesterone levels, which leads to a reduced rate of GVBD and is important for embryonic development [87].

The transient receptor potential cation channel, subfamily M, member 7 (TRPM7), is a protein that is regulated by EGF and inositol-trisphosphate 3-kinase A (ITPKA). TRPM7, which interacts with calmodulin 2, downstream of EGF, was found for cleavage-stage embryo prediction [88,89]. It has been reported that cleavage and blastocyst formation rates were decreased by DOs compared to intact COCs, suggesting that the reduced first cleavage affected the reduced blastocyst formation rate for DOs, but not the subsequent embryo development in bovine samples [83].

Due to the size of follicle development, the small follicles of embryos in heterogeneous cycles had a possibility of failure to reach blastocysts than the large follicles of embryos in homogeneous cycles [90]. CC genes correlated to blastocyst development, and HAS2 and GDF9 represented a high expression in CCs from oocytes that developed better-quality blastocysts [85]. Embryonic development relates to the expression of three genes of GDF9 activity (HAS2, PTGS2, and GREM1). The expression of these cumulus genes shows a high expression in oocytes that progressed into high-quality embryos (grades 3, 4, and 5) compared to low-quality embryos (grades 1 and 2). This confirms that the morphology of human embryos after 3 days of culture depends on the expression of these cumulus genes [90].

Another study of gene expression analysis in humans using the reverse transcriptase-polymerase chain reaction (RT-PCR) found that GREM1 and HAS2 represent high expression, but not PTX3 in high-quality human embryos [13]. A higher expression of PTX3 was found in human COCs in the normal development of embryos on Day 3 compared to oocytes that fail in fertilization [91]. BMP2 has functions in the formation of the oocyte and other follicular cells in hamster fetal ovaries in vitro. Moreover, it organized meiosis and anti-apoptosis on germ cells. Bone morphogenetic protein receptor-2 (BMPR2) was very important for FSH-mediated follicular development in the pre-ovulatory period in human GCs. The result of this analysis of human CCs indicates significant changes in BMP2 protein expression related to the quality of oocytes and embryos [92]. The expression of BMP2 in human CCs is very important for revealing the molecular pathway taken by quality oocytes and embryos.

Embryo quality can also be predicted using apoptotic-related genes. Survivin, the smallest member of the inhibitor of the apoptosis protein family (IAP) gene family, is located on chromosome 17q25 and encodes a 142 amino acid protein 9. Expression of surviving has been described during embryonic development and in several proliferating normal adult tissues such as skin, endometrium, and granulosa cells. In pig and mouse zygotes, the accumulation of survivin occurred at chromosomes in the metaphase and at the spindle midzone in the anaphase and telophase during the first cleavage. In early mouse embryos, the failure of cleavage resulted from a loss of survivin activity. Survivin plays a role in regulating early embryo development in several processes in cell division [93]. A previous report demonstrated that survivin expression depends on LH and FSH gonadotropins [36]. The survivin expression was regulated by FSH through the phosphatidylinositol-3-kinase/AKT (PI3K/AKT) pathway, while LH was regulated by FSH through the extracellular signal-regulated kinase1/2 (ERK1/2) pathway. The main functions of cell survival are the modulation of cell cycle, cell survival, and cell death. As a member of IAP, survivin can suppress apoptosis via inhibition of caspase-3 and -7 [12], which are apoptotic caspases [20].

It has been reported that LH and FSH enhanced caspase-3 and -7 [21]. Caspase-3, the primary executioner of apoptosis, plays a critical role during apoptosis, including chromatin condensation, DNA fragmentation, nuclear envelop breakdown (NEBD), plasma membrane blebbing, cell disassembly, and the formation of the apoptotic body [20]. On the contrary, caspase-7 plays an important role in the demolition phase of apoptosis, generating reactive oxygen species (ROS), and detaching cells from the ECM. Therefore, the expression of survivin, caspase-3, and caspase-7 could be used as genetic biomarkers for the evaluation of oocytes and embryos under an ART program in CCs of PCOS patients [36].

It has been reported that AMHR2 and LIF showed significant expression differences between high-quality and low-quality embryos [73]. AMHR2 revealed a negative correlation with embryo quality, whereas LIF CC expression represented significant differences between the CCs of high- and low-quality embryos. Interestingly, when both AMHR2 and LIF expressed low, there was a high possibility of the development of high-quality embryos. When AMHR2 expressed high and LIF expressed low, there was a high possibility of the development of low-quality embryos. With all other combinations of AMHR2 and LIF expressions, the development of high-quality or low-quality embryos was equally possible. In this report, AMHR2 combined with LIF demonstrated a high predictive power for estimating the quality of embryos [94]. A set of highly predictive genes would probably result in a good prediction model, where the decision tree model seems to have high clinical applicability [94]. Alterations in CCs and oocyte communication in folliculogenesis may be subsequently responsible for the poor quality of embryonic development.

## 7. Cumulus Cell Biomarkers and Pregnancy Outcome

CCs may represent a material for the non-invasive assessment of embryo selection and its potential to result in pregnancy, which are discarded during IVF or ICSI procedures. Fertilization and embryonic development markers would increase the chance of pregnancy by the selection of optimizing oocytes and embryos. Several studies have reported that some genes, expressed in cumulus cells, may be useful as potential biomarkers of pregnancy outcome [95]. It has been demonstrated that a significant enhancement of implantation and pregnancy rates was performed by the potency of cumulus-aided embryo transfer, using autologous cumulus cells [75]. Thus, CCs may play an important role in embryonic development and can provide an improvement in the embryo–uterine adhesion due to both physical proximity and the secretion of cytokines, and several growth factors are required to aid the implantation process [96].

Long noncoding RNAs (lncRNAs), longer than 200 nucleotides, are a new class of transcripts. The lncRNA could regulate essential pathways that contribute to fertilization, and development of the embryo. In human CCs, the relative expression levels of AK124742 and PSMD6 in the pregnancy group were significantly higher than in the nonpregnancy group. AK124742 is a newly detected lncRNA that was identified as being naturally antisense to PSMD6. AK124742 and PSMD6 expression was correlated with embryo quality and clinical pregnancy outcome. Thus, AK124742 and PSMD6, as a new lncRNA–mRNA gene pair, may be considered as new potential biomarkers for embryo selection [97].

Several genes involved in modulating the cumulus matrix function and expansion might be indicative of oocyte developmental competence and pregnancy, such as VCAN (89), PTX3, and PTGS2. VCAN, an ECM proteoglycan, cross-links HA in the matrix of expanded CCs and stabilizes HA in pericellular matrices [22]. The product of the PTX3 gene is another ECM protein that interacts with HA in the matrix of expanded CCs [29,30,53]. Additionally, it has been reported that the pregnancy prediction depends on the expression of SDC4 and VCAN [89,98]. In humans, a correlation was found between the cumulus gene and pregnancy, and the expression of VCAN and PTGS2 was significantly higher in CCs from oocytes, yielding a pregnancy resulting in a live birth. The expression of VCAN, GREM1, and phosphofructokinase platelet (PFKP) in CCs correlated with birth weight in patients at 38 weeks of gestation. Thus, in human CCs, PTGS2, VCAN, PFKP, and GREM1 expression may identify in the oocytes with a high potential for development, leading to enhanced implantation rates and a higher developmental capacity throughout gestation [91]. Moreover, it has been shown that the expression levels of GDF9 and BMP15 mRNA in a pregnant group were significantly higher than those in a non-pregnant group [56]. Therefore, GDF9 and BMP15 mRNA expression can be used as indicators to predict clinical pregnancy outcomes [47]. GDF9 and its downstream (HAS2 and PTGS2) may be correlated with embryo quality and positive pregnancy. Increased levels of GDF9 and BMP15 expression are associated with positive pregnancy as well as with fertilization rate and embryo quality. Among OSFs, GDF9 is an important candidate factor, as its downstream genes HAS2, cyclooxygenase 2, GREM1, and PTX3 were significantly increased in CCs surrounding high-quality oocytes [11].

In human CCs, the gene expression of the PI3K/AKT pathway can be used as a predictive marker for successful embryo implantation. Akt1, Bc1211, and Shc1 in the PI3k/AKT pathway were related to oocyte maturation and ability. The PI3K signaling pathway was a key regulator of COC function and regulated the maintenance or activation of oocytes and the proliferation, differentiation, and stress response of granulocytes [99]. AKT1, a member of the serine/threonine-protein kinase family, is involved in the regulation of many cellular processes, including metabolism, proliferation, cell survival, growth, and angiogenesis. ARHGEF7, CCND1, E2F1, HRAS, and SSP1 were also included in the participation of proliferation control [100].

HAS2, secreted by the oocyte, is considered to be a marker of human pregnancy or live birth [89,90]. Expression of diaphanous-related formin 2 (DIAPH2), involved in spindle dynamics and nibrin (NBN) as well as in chromosomal alignment, was significantly higher in embryos resulting in implantation and clinical pregnancies as well as live birth. In humans, pronuclear fading together with expression of the DIAPH2 gene were independent prognostic factors of clinical pregnancy rate and live birth [101]. It has been found that CAMK1D and EFNB2 could help in selecting the embryos to transfer with the highest chance of pregnancy [88]. EFNB2 is a transmembrane protein that belongs to the largest subfamily of receptor protein-tyrosine kinases. This protein mediates the developmental events, especially in the nervous system and in erythropoiesis. In human MGCs, EFNB2 expression was described mainly during luteinization.

CAMK1D encodes a member of the Ca-/calmodulin-dependent protein kinase 1 subfamily of serine/threonine kinases. Vascular endothelial growth factor (VEGF) increases the EFNB2 expression in endothelial cells. The higher expression of EFNB2 in CCs might be a higher VEGF content reflection in the follicles and better vascularization of the follicles. Therefore, a higher VEGF content in follicular fluid correlated with the perifollicular vascularity grade, and a higher vascularization resulted in higher rates of fertilization, embryos, and pregnancy [58].

Apoptosis in MGCs and CCs can be induced by oxidative stress, hyper-androgenemia, and a disturbance of gonadotropin hormones. Apoptotic cells were significantly lower in pregnant women than those who did not become pregnant [36]. Apoptosis-related genes are involved in poor oocyte and embryo development and impaired blastocyst development. CC apoptosis is related to embryo quality and pregnancy rates. Therefore, these genes in CCs could be reliable biomarkers for oocyte and embryo selection and related to the poor quality of the corresponding embryos [36]. Survivin (an anti-apoptotic gene), caspase-3, and caspase-7 (two pro-apoptotic genes) have been reported to be involved in the chance of pregnancy. In CCs of women with polycystic ovary syndrome (PCOS), the survivin gene expression was lower, while caspase-3 and -7 were higher. Therefore, the degree of granulocyte apoptosis might be inversely related to the developmental capacity of oocytes. CC apoptosis may be related to the pregnancy and live birth of related embryos in ART treatment. The proportion of apoptotic CCs is closely related to the outcome of intracytoplasmic sperm injection (ICSI). It can be considered as a predictor of pregnancy and live birth, and the percentage of apoptotic CCs is an independent prognostic factor for these outcomes. For every 1% increase in CC apoptosis, the clinical pregnancy and live birth rates will decrease by 11–12% [102].

Oxidative stress induces DNA damage and initiates apoptosis. Its markers include reactive oxygen species (ROS), lipid peroxides [31], total antioxidant capacity (TAC), and 8-hydroxy-2′-deoxyguanosine. Studies have shown that an increase in ROS was associated with decreased fertilization capacity, embryo quality, and oocyte implantation capacity [10]. An intrafollicular ROS leads to lack of antioxidant defenses in advanced maternal age and alters the function of mitochondrial CCs during the development of oocytes through free radical formation or reduced ATP production [1]. Since mitochondria are important in supplying embryonic energy, the quantitation of mtDNA can be used as an implantation biomarker. The comparison between implanted and non-implanted groups found that the mean mtDNA content was significantly higher in CCs surrounding oocytes and successfully implanted human embryos. For human embryos of equivalent quality, this amount of mtDNA is related to the potential of embryonic implantation [31]. Therefore, the quantitation of mtDNA of CCs is an oocyte competence biomarker and a guide to selecting embryos to transfer for a successful pregnancy during IVF procedures.

## 8. Summary

CCs are different from undifferentiated granulosa cells. Gonadotropins (FSH and LH) stimulate CC differentiation, and the lineage of CCs is specified by the actions of OSFs (GDF9 and BMP15). These two oocyte-specific growth factors bind to their receptors to activate the SMAD2/3 pathway and induce the cumulus gene expression profile, which includes the suppression of LHR and progesterone receptor expression, preventing CCs from responding to key endocrine stimuli of MGCs. Meanwhile, the signals of GDF9 and BMP15 activate the key cumulus-specific genes expression, such as glycolysis enzymes and cholesterol synthetic pathway enzymes that are essential mediators of healthy oocyte function. CCs are in direct contact with the oocyte through gap junction connections with the oocyte plasma membrane. Characteristic of CCs is the higher rate of proliferation, higher expression of AMH, lower capacity of steroidogenic, and lower expression of LHR. CCs can secrete HA for COC expansion.

During oocyte maturation, the expansion of the COC is important for meiotic maturation by rupture of the gap junction, which leads to decreased cAMP in the oocyte. In response to the LH surge, COCs undergo an expansion and synthesis of HA by CCs. This process requires the synthesis of an HA-rich matrix and factors that bind HA to stabilize the matrix. Cumulus expansion requires stimulation by a ligand, FSH, or EGF-like peptides and the activation of the MAPK3/1 and MAPK14 kinase signaling pathways in CCs and OSFs. OSFs secrete growth factors (GDF9 and BMP15) and activate the SMAD2/3 pathway to induce the expression of several genes, including HAS2, cyclooxygenase 2 (COX2; PTGS2), GREM1, and STAR, and to repress LHR. CCs express members of the Toll-like receptor (TLR) superfamily that can respond to specific ligands (matrix-derived or pathogen-derived), leading to the activation of innate immune-related genes and inflammation. These genes include IL6, PTGS2, TNFA, TNFAIP6, and PDCD1. PGE2 synthesized by PTGS2, IL6, and TNFα and other cytokines and chemokines are released from CCs. The degradation of polymeric HA by hyaluronidases is presumed to lead to the generation of HA fragments that activate TLR2 and TLR4.

Based on previous studies, it is believed that CCs can help predict oocyte quality, fertilization, and early embryonic development and pregnancy. During fertilization, CCs secrete many factors to help sperm fertilization. PGE1, PGE2, PGF2, PSSS35, PACAP, and PTX3 are produced by CCs to help sperm motility, acrosome reaction, and fertilization. The knockout of these genes leads to a reduced rate of fertilization in humans and many other mammalian species. Moreover, CCs are helpful for cleavage and blastocyst development. A lack of CCs before IVF reduces the rate of cleavage and embryonic development. TRPM7 and ITPKA genes are used to predict the good quality of embryonic development at the cleavage stage. In addition, it seems that HAS2, GREM1, and GDF9 are related to the development of morula and blastocysts. In contrast, apoptotic and anti-apoptotic genes in CCs, such as caspase and survivin, are used to predict the quality of oocytes and embryo development. Furthermore, it has been indicated that several genes are involved in modulating cumulus matrix function and expansion (VCAN, PTX3, and PTGS2), apoptosis-related genes (survivin, caspase-3, and caspase-7), and ROS genes, which are indicative of the competence of oocyte and the subsequent embryonic development and pregnancy. Current studies have provided a deeper understanding of CC function and its factor to contribute to oocyte maturation and fertilization. For lncRNA to predict embryo selection and pregnancy, few studies have been reported and are still unclear. It is also critical to investigate lncRNA related to pregnancy. With numerous women attempting an infertility treatment, CC factors and invasive techniques are important to study and useful for a successful ART. This review provides an insight into the mechanism of CCs in various stages of development, as well as potential areas to explore further to gain a better understanding of the intrafollicular environment related to oocyte developmental competencies.

## Figures and Tables

**Figure 1 cells-10-02292-f001:**
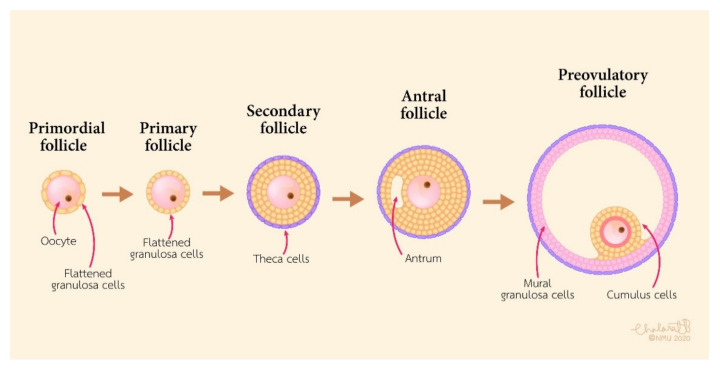
Schematic diagram of follicular development.

**Figure 2 cells-10-02292-f002:**
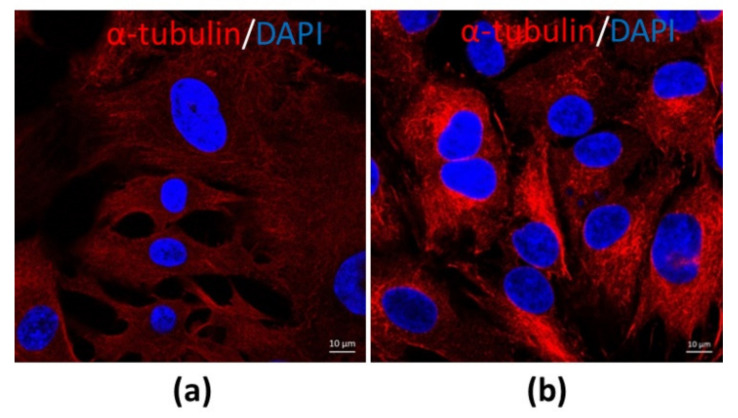
The morphological differences between MGCs and CCs. (**a**) MGCs at 63× magnification; (**b**) CCs at 63× magnification. The cells were stained by the immunofluorescence method (unpublished data).

**Figure 3 cells-10-02292-f003:**
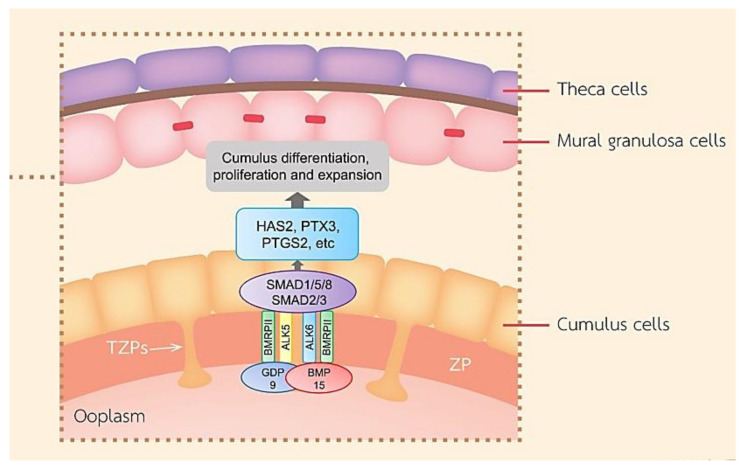
A schematic diagram exhibiting roles of the main TGF-β superfamily member proteins in regulating ovarian function. BMP-15 and GDF-9 promote cumulus marker transcripts. GDF-9 and BMP-15 are important for the function of GCs, CC differentiation and proliferation, COC expansion, and hyaluronan production leading to ovulation.

**Figure 4 cells-10-02292-f004:**
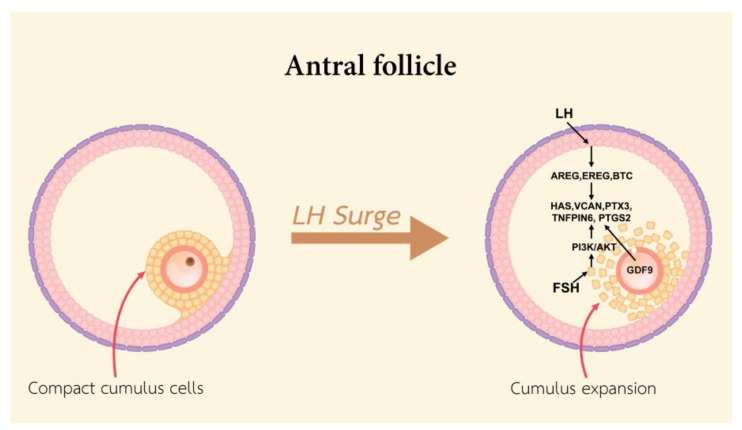
A schematic diagram exhibiting the functional roles of LH and FSH in stimulating cumulus expansion.

## Data Availability

Not applicable.

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
