# Peer review of "The Function of Cumulus Cells in Oocyte Growth and Maturation and in Subsequent Ovulation and Fertilization"

_cells, 2021, doi:10.3390/cells10092292_

Round 1
Reviewer 1 Report
This is an exceptionally written paper, but in my personal opinion, a systematic review would be more robust through its methodology.
Reviewer 2 Report
The manuscript has been improved as requested by the reviewer, after the first submission.
Reviewer 3 Report
The manuscript is improved and the review of latest knowledge about the role of cumulus cells are important and it maybe directed to further editorial processing.
This manuscript is a resubmission of an earlier submission. The following is a list of the peer review reports and author responses from that submission.
Round 1
Reviewer 1 Report
The paper has been adequately improved in terms of language and grammar. What it lacks is clear structure, methodology and synchonisation of reporting.
Author Response
Thanks for the comments and suggestions. After further discussion on the comments and suggestions, we believe that the structure of this review is clearly built. We also believe that it is relatively well organized on what we want to say as the title indicated.
Bongkoch Turathum

Reviewer 2 Report
The manuscript by Turathum et al. entitled "The Function of Cumulus Cells in Oocyte Growth and Maturation and in Subsequent Ovulation and Fertilization” is a review on cumulus cells functions covering the major points of the interaction with oocyte.
Based on these data, the present study is on a topic of relevance and general interest to the readers of the journal. I suggest after the first revision to publish as it stands as the changes made were satisfying.
Author Response
Appreciated for the positive comments.
Reviewer 3 Report
The main reason of the review was presented in the statement: ““Therefore, understanding the function of CCs during follicular development may be helpful for predicting oocyte quality and subsequent embryonic development competence as well as pregnancy outcomes in the field of reproductive medicine and assisted reproductive technology (ART) for infertility treatment. “
It is interesting however very general without pointing out the most important/ original part of the recent research.
The topic was and is under research for many years and still new facts, interactions, genes and other factors are discovered. Authors pointed out the potential usefulness of the newest facts for assisted reproductive technology for infertility treatment what is obviously very important.
The originality of the topic: The manuscript is divided into 7 logical chapters: Introduction a/definition of cumulus cells; b/differentiation of granulosa cells; 2.Interaction of oocyte and cumulus cells; 3.Mechanisms of cumulus expansion and ovulation; 4. Cumulus cells in fertilization; 5.Cumulus cells in embryonic development; 6. Cumulus cells biomarkers and pregnancy outcome; 7. Summary .
This division is clear and easy to read. The Authors included their own results which were not published so it is new addition compare to other published material. Recently many other reviews and research papers considering the role of the cumulus cells were published , several of them were cited by Authors.
The summary is consistent with the evidences presented in the manuscript, there is lack of the arguments for and against, Authors used the formula: enumeration of facts in the chapters.
They addressed the main reason of the writing the manuscript, however the clear conclusion will be helpful and make the manuscript more novel.
The main reason of the review was presented in the statement: ““Therefore, understanding the function of CCs during follicular development may be helpful for predicting oocyte quality and subsequent embryonic development competence as well as pregnancy outcomes in the field of reproductive medicine and assisted reproductive technology (ART) for infertility treatment. “
Author Response
Thanks for the comments and suggestions. The purpose of this review is to understand the functional role of CCs during follicular development. To predict oocyte quality and subsequent embryonic development competence and pregnancy outcomes are not the key points in this review as they are only potential factors of CCs because many of them are under development and unconfirmed as of yet. Therefore, we did not list all recent reports.
We choose to summarize without inclining to any arguments. As mentioned above, the main purpose of this review is to describe the differences between CCs and MGCs and the functional role of CCs during follicular development. Upon understanding the functional role, it can be helpful in predicting oocyte quality and subsequent embryonic development competence as well as pregnancy outcomes, in which technologies still is under development and not certainly confirmed yet.
Best Regards,
Bongkoch Turathum
Round 2
Reviewer 1 Report
My initial comments still stand.
Reviewer 3 Report
The "revised" version of the manuscript is almost exactly the same as the original version no.1. Also, the answers from the Authors are not satisfactory.
In this situation I am not changing my first opinion.